# Advanced Innovation Technology of BIM in a Circular Economy

Marcel Behún and Annamária Behúnová *

Faculty of Mining, Ecology, Process Control and Geotechnologies, Institute of Earth Resources,
Technical University of Kosice, Park Komenského 19, 042 00 Kosice, Slovakia; marcel.behun@tuke.sk
* Correspondence: annamaria.behunova@tuke.sk

**Abstract:** The traditional concept of the primary, secondary, tertiary and later quaternary economy is based on several structurally divided and related tasks and processes in processing raw materials and earth resources. Gradually, a new concept of the functioning of the economy was created, called "circular economy" or "circular economy". Its basis is the transformation of linear economic processes managing the use of raw materials to create a sustainable economic growth model. The circular economy transforms economic activity associated with the consumption of limited resources into the more efficient reuse of resources. Based on the above, the presented article aims, based on theoretical and empirical analysis, to identify the potential of processing and using non-energy raw material—recycled aggregate—in the construction industry and to propose a concept for information modeling of the parameters of sustainable construction using this non-energy raw material per the principles of the circular economy. The solution to this research problem is realized through theoretical analysis and comparison of approaches to the circular economy, reuse of non-energy raw materials in the construction industry and analysis for the creation of a concept based on the use of information needed for sustainable construction planning through building information modeling (BIM). Based on my research, my results will be presented, the applicability of which is verified through a case study. The object of the case study is the construction of a new building, which will represent a set of five similar constructions interconnected by underground floors (garages, technical facilities of buildings) and communication spaces (corridor, hall). The priority of the construction of the centre is to build a sustainable building, i.e., to implement the work using sustainable methods with the greatest possible use of sustainable materials and procedures, which will reduce the impact on the ecosystem and support the goals of the circular economy. Traditional, natural raw materials will be replaced by recycled secondary raw materials within individual constructions and elements. When choosing suitable raw materials, the design of the BIM library of sustainable elements will help. The BIM library will act as a link between manufacturers and BIM digital replicas of real building products and components.

**Keywords:** BIM; circular economy; sustainable material

## 1. Introduction

Environmental problems such as loss of biodiversity, water, air and soil pollution, resource depletion and overexploitation of land are increasingly threatening life support systems on Earth [1]. Social expectations are not being fulfilled due to problems such as high unemployment, poor working conditions, social vulnerability, the poverty trap, intergenerational equality and deepening inequalities [2]. Economic challenges, such as resource supply risk, problematic ownership structures, unregulated markets and faulty incentive structures, lead to increasingly frequent financial and economic instabilities for individual companies and entire economies [3,4]. To address these and other sustainability issues, the concept of the circular economy has recently been developed in national policy programs [5,6]. The introduction of the concept of sustainability can be traced back to the growing evidence of global environmental risks such as ozone depletion, climate change,

loss of biodiversity or changes in the nitrogen cycle. These risks have been systematically investigated since the 1960s, which raises questions as to whether it is possible to maintain the current trends of prosperity in the future [1]. More than 100 definitions of circular economy are used in scientific literature and professional journals. So many different definitions are used because the concept is used by a diverse group of researchers and practitioners [6,7]. Scholars emphasize a different aspect of this concept than financial analysts. The diversity of definitions also makes it difficult to measure circulation. Definitions often focus on the use of raw materials or system change.

Definitions that focus on resource utilization often follow the 3R approach:

- reduce—reduce (minimum use of raw materials),
- reuse—reuse (maximum reuse of products and components),
- recycle—to recycle (high-quality reuse of raw materials).

The use of resources is minimized (reduced). Reuse of products and parts is maximized (reuse). And last but not least, the raw materials are reused (recycled) with high quality (Figure 1). This can be achieved by using goods from multiple people. Products can also be converted into services. For example, Spotify sells listening licenses instead of CDs. In this system, value is created by focusing on value preservation [8].

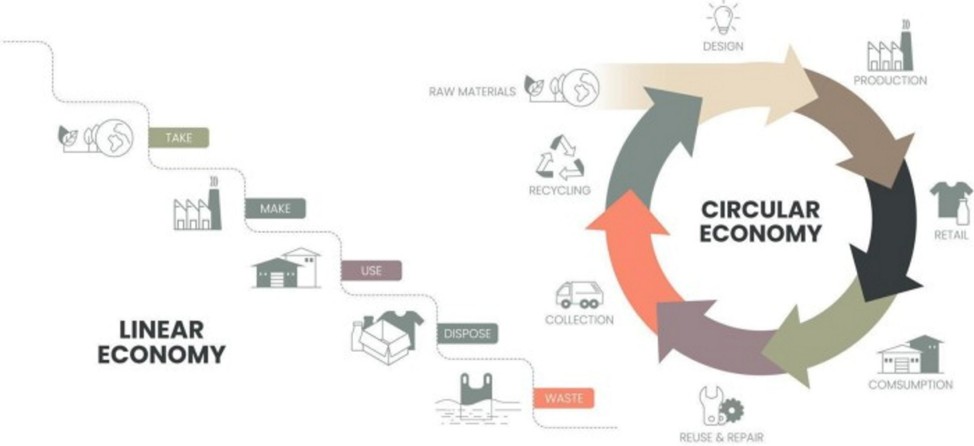

**Figure 1.** Linear vs. circular economy [9].

The circular economy is fundamentally different from the linear economy. In a linear economy, we extract raw materials, process them into products and throw them away after use. In the circular economy, we close the cycles of all these raw materials. Closing these cycles requires much more than just recycling. It changes how value is created and preserved, how production is more sustainable, and what business models are used. A circular system and a linear system differ in how value is created or maintained. A linear economy is traditionally guided by a gradual plan, i.e., step by step [10]. The view of sustainability is different in the circular economy than in the linear economy. When working on sustainability within the linear economy, we focus on ecological efficiency, which means that we try to minimize the environmental impact to achieve the same output. This will increase the period during which the system will be overloaded [11,12].

Within the circular economy, sustainability is sought by increasing the eco-efficiency of the system. This means that the ecological impact is minimized, and the ecological, economic and social impact is even positive [13]. When we focus on eco-efficiency to create a positive impact, we use it to strengthen ecological, economic and social systems. To achieve eco-efficiency, residual streams must be reused for a function that is the same (functional recycling) or even higher (upcycling) than the material's original function. As a result, the value is fully preserved or even increased. In a linear economy, it is different.

An eco-efficient system usually works on downcycling: a product (part) is reused for a low-quality application that reduces the value of the material and makes it difficult to reuse the material stream [14–16].

Mineral raw materials are the building blocks of many products, which creates a problem in the long term due to the non-renewability of extracted raw materials. Therefore, when introducing a circular economy, changes in production and consumption are necessary, which also considers the monitoring framework that monitors the supply of critical mineral raw materials. Especially nowadays, when the EU countries are increasingly aware of their dependence on the import of mineral raw materials from other parts of the world and with geopolitical tension, the security of the supply of mineral raw materials is an integral part of the security of the EU countries as well as the original countries. We are in a period when the concept of the circular economy (CE) and the closely related so-called critical mineral resources. The supply of these raw materials is exposed to a high risk. They are of great economic importance, as reliable and unrestricted access to them is important for European industry and value chains. The list of these raw materials was last reviewed in 2020 when it contained 30 raw materials and is reviewed regularly every three years (e.g., antimony, barite, bauxite, lithium, magnesium, niobium and others) [17–20].

The extraction of non-energy raw materials provides several basic raw materials for European production and construction activities. In 2007, it achieved a turnover of approximately EUR 49 billion and secured jobs for about 287,000 people. However, the economic importance of this sector is much greater if the added value for the larger processing industries, whose activity depends on uninterrupted supplies of mineral raw materials, is taken into account. In November 2008, the European Commission adopted an initiative on raw materials, establishing targeted measures to ensure and improve access to raw materials within the EU and globally. Within the EU, a group of factors has been defined that could affect the competitiveness of the non-energy raw material extraction sector. The European mining industry for non-energy raw materials is often divided into three sub-sectors. This division is based on the physical and chemical properties of the relevant mineral raw materials and, above all, on their use and takes into account the supplied processing industries [20–22].

Three main groups of minerals, according to the sectors of extraction of non-energy raw materials [20,21]:

- *construction raw materials*—Construction minerals usually include aggregates made up of different-sized particles, such as sand, gravel and various types of crushed rock (e.g., chalk, limestone, sandstone, slate, etc.), natural rocks (e.g., marble or granite), as well as a whole range of clays, gypsum or clay-slate. The extraction of construction minerals, especially stone (crushed rock, sand, gravel and aggregate), represents the largest subsector of the extraction of non-energy raw materials in the EU in terms of value and volume. Potential sources of construction mineral raw materials are abundant in all Member States and are mined in large quantities (approximately 3 billion tons per year). A small (but increasing) amount of aggregate is also obtained from by-products of other industrial processes, for example, arc furnaces, residues from mineral processes, such as kaolin sands, and from surface mining of stone, as well as from the repeated processing of materials used in construction. However, the amount mined varies greatly between countries: most construction minerals are mined in Germany, France, Italy, Spain and the United Kingdom. Aggregate has a wide range of uses, including the construction of buildings, roads and railways. The demand for aggregates is, therefore, closely related to the level of construction of new buildings, the maintenance and repairs of existing buildings and the scope of civil engineering projects. Around 22,000 sites are estimated to be mined across the EU, many of which are close to construction areas. In the Netherlands and Belgium, raw materials are often transported over long distances on navigable rivers and canals due to relatively limited aggregate stocks. Similarly, densely populated cities such as London and Paris have to import most of the aggregate from more distant locations.

The bulk of the aggregate price comprises transport costs, meaning most markets are local or regional, and international trade is relatively small. Aggregate mining requires an adequate network of surface mines and quarries to shorten transport distances, reduce related costs and reduce the impact on the environment;

- *industrial minerals*—Minerals form ore and non-ore raw materials. Industrial minerals can be loosely defined as physical minerals (e.g., bentonite, borates, calcium carbonate, diatomites, feldspar, kaolin, plastic clays, quartz and talc) or chemical minerals (e.g., salt, potash and sulphur). Various industrial minerals are mined in the EU, including feldspar, kaolin, magnesite, perlite, potash and salt. While some are found in almost half of the Member States, others (such as fluorite, mica, mineral phosphate and sulfur) are mined in only one or two countries. In the last ten years, the extraction of most industrial minerals has been maintained at a stable level or is growing. Industrial minerals are used in large industries, but unlike base or rare metal minerals, they are not marketed or sold as standardized products through centralized markets. Instead, they are most often sold directly to end users. Although a few industrial minerals are traded worldwide, most are processed and used in production within the EU. The delivery price is significantly affected by relatively high transport costs, which also effectively limit the geographical availability of suitable resources;
- *metallic minerals*—Metallic minerals include a wide range of ores from which metals or metallic substances are extracted by processing, for example, bauxite, chromium, copper, gold, lithium, manganese, nickel, selenium, silver, tin, tungsten, etc. Relatively few metallic minerals are mined in the EU. These include chromium, copper, iron ore, nickel, lead, silver and zinc. The geology of the European continent is such that other metallic minerals are not found in the EU territory in large quantities or are in places where mining is technically difficult and expensive. For this reason, mines are currently located in only a small number of Member States. Only Austria, Finland, Greece, Ireland, Poland, Portugal and Sweden account for over 1% of the global mining of a particular metallic mineral. Thus, many metallic minerals are imported from other regions of the world. The most important primary outlet for metal ores and concentrates in the EU is the refining and processing industry, which produces semi-finished and finished products for many other product segments. Regarding recycling, the construction minerals sector is the most problematic of the three subsectors, given the volume used, while metals provide the most economic opportunities for recycling. Many metals, including iron and steel, copper, tin, lead and aluminium, are relatively easy to recycle as they can be melted down and reworked without losing their significant characteristics. This potential is not yet fully utilized, as products at the end of their life cycle are often exported outside the EU and thus subsequently lost to the European market.

*BIM and Circular Economy*

Building information modeling is an intelligent process for creating and managing construction projects. BIM brings many advantages, especially in project design, construction and use of the building. Building information modeling tools can be used in individual phases of the life cycle of buildings. Through them, it is possible to manage projects faster, more economically and with a lower environmental impact. Implementing BIM in individual processes and works increases work productivity, leads to time savings, eliminates errors, controls the entire project, increases competitiveness and increases the overall profitability of the project [23–25].

The beginnings of BIM development are considered to be from the 1970s when American professor Charles M. Eastman from the Georgia Institute of Technology first published in the AIA Journal a workflow prototype called BDS (building description system), which described interactive elements and operations whose changes it would be enough to convert only once, and then their changes would also occur on subsequent elements or drawings. With this, he practically described BIM as we know it today and identified all the basic

problems in architectural design for the next 50 years. His last project from 1977, GLIDE (Graphical Language for Interactive Description System), contains the features and characteristics of modern BIM procedures. This concept reached Europe in the early 1980s through studies and attempts at commercial use. In Europe, it was used for the first time in history through the RUCAPS (Really Universal Computer-Aided Production System) software for constructing prefabricated components in 1986 during the renovation of London Heathrow Airport. The Hungarian programming genius Gábor Bojár is responsible for a bizarre part of BIM development. They illegally smuggled Apple computers through the Iron Curtain for his wife's jewellery to develop the program known today as ArchiCAD. ArchiCAD was also the first BIM software accessible on personal computers in 1987 [26,27].

BIM had seen the greatest progress and development in the last decade when it went from theoretical concepts and individual attempts to mass commercial use. It is used throughout the construction process, from design to construction completion, and at the same time during the building's life cycle for its management, maintenance and modernization. Its development and standardization were mainly contributed by its introduction at the state level in countries such as Finland, Norway and Denmark, where the introduction of BIM is mandatory for state-funded construction projects, the cooperation of construction and technology companies that joined together to create standards and procedures for implementing BIM into the construction process and also the creation of databases for manufacturers of construction components, where they can upload models that are available to the public completely free of charge [26,27].

In recent years, we have seen a growing trend towards increasing the use of BIM in practice. The main reasons for the gradual implementation of BIM are the development of new software tools that offer countless new and innovative functions, making it easier for the user to plan, create and manage information models. The priority of BIM implementation is the use of unique, innovative tools in the management of public property and, at the same time, expanding the possibilities of evaluating proposals considering the sustainability of construction and the reduction of carbon emissions. Significant public organizations, national authorities and governments also greatly influence the development and increasing implementation rate. Currently, some countries already require BIM tools for public contracts planning, development, management, implementation, and management of individual projects [28,29].

Builders and academics have made a concerted effort to introduce circular economy concepts to the construction sector. Based on several studies, the circular economy in construction can be understood as a new model for maintaining the value of resources and preventing the use of original materials and waste outputs, not only by recycling and reusing materials but, above all, by reducing the need for resources [30–32]. The definition is consistent with our previous statement: "Circular economy through a prioritized order of Reduce, Reuse and Recycle (3Rs)". The main idea is that not only the material flows should be maintained in cycles, but also the buildings should have a long life so that the functionality is not destroyed. In other words, the possibility of extending the life of buildings should be prioritized (Figure 2). Then strategies for restoring material values, such as material recycling and reuse, will be considered when the building's functions are eventually lost [33].

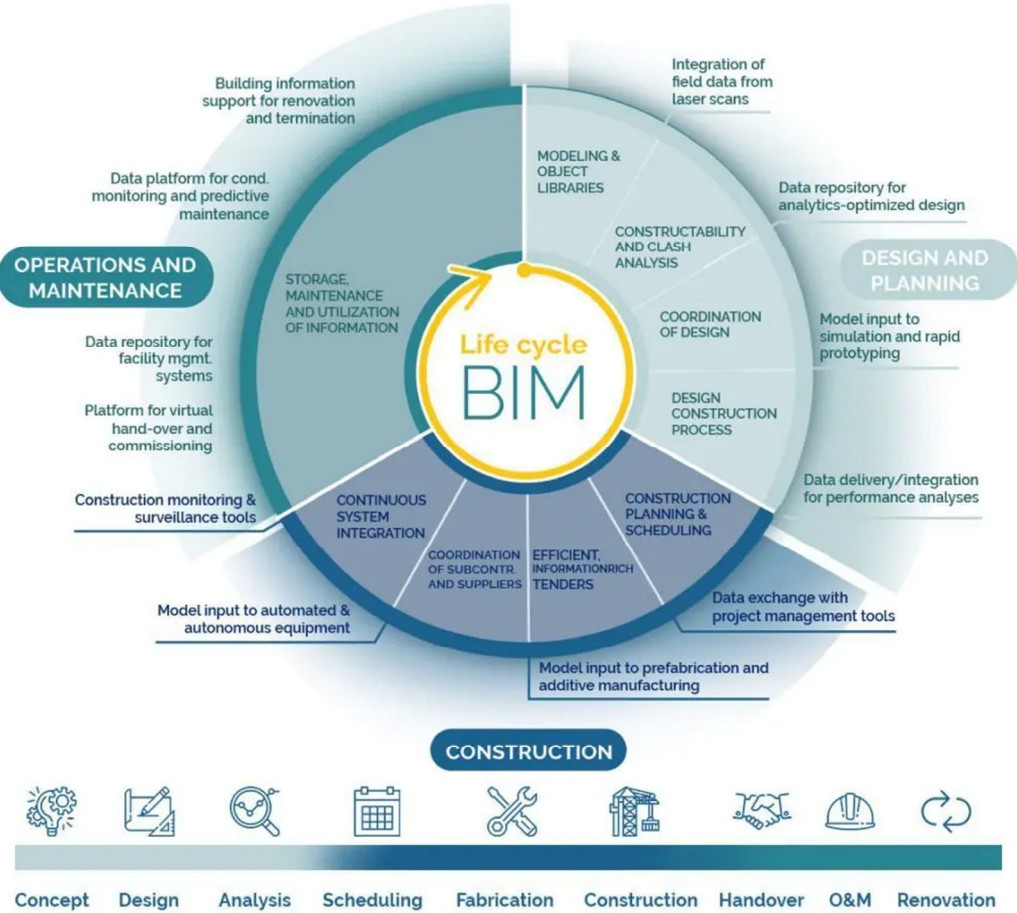

**Figure 2.** Life cycle BIM [34].

## 2. Materials and Methods

The starting point for adopting a circular construction approach is understanding the construction supply chain and how its needs and challenges match the potential of a circular construction approach. Based on the supply chain operations reference (SCOR) model, the construction supply chain process consists of multiple functions throughout the project life cycle. The entire life cycle includes the phases of design, production, construction (or installation), maintenance, commissioning and end of life (deconstruction or demolition) [35]. The entire life cycle of buildings requires multiple project participants, from the design phase to the end-of-life phase. In addition to policy/legislators and investors, the main project participants involved in a cyclical supply chain network include project owners, architects and engineers, construction contractors and subcontractors, facility managers, suppliers and distributors, and recycling plants. The supply chain process begins with creating design intents from building owners, architects and engineers. Then the design information is sent to the suppliers to coordinate shop drawings and material production. After delivery and quality control, the materials are installed on-site and eventually enter the "End of Life (EoL)" phase. In the case of a circular structure, the building is deconstructed at EoL [19,36,37].

The rate of use of aggregate from recycled material (RCA—recycled concrete aggregate) is increasing, and its application is increasing. The main disadvantage and obstacle to increasing the rate of recycled concrete is its unstable properties. Despite the mentioned fact, its degree of application is diverse: through the filling of sewers, for the creation of backfills, underfills of parking lots or floors, embankments of road bodies, material for the construction of anti-flood dams, foundation and drainage layers or even as a substitute for natural stone for concrete (mainly larger fractions). The use of recycled concrete varies

depending on the environment. Table 1 shows the options and recommendations for using recycled concrete aggregates at individual levels of the environment. Aggregate from recycled high-quality concrete waste with suitable geometric, chemical and physical properties can substitute small fractions in concretes of classes C16/20 to C40/55 and for environments X0 and XC. Grains up to approx. 4 mm can be used as screed substitutes. Currently, the use of recycled concrete to remove phosphates from wastewater is being investigated. The aggregate fraction of 0.125–0.250 mm has proven to be an excellent absorbent because it exhibits high absorbency. Research confirms that the adsorption capacity of aggregate PO43 is 0.006–0.134 g/g. Absorption capacity shows the amount of substance that is superscribed in an equilibrium state per 1 g of aggregate [38,39].

**Table 1.** Possibilities and recommendations for the use of recycled concrete aggregate [38,39].

| Description of the Environment | Informative Examples of the Occurrence of the Degree of Environmental Impact | Usability of Recycled Concrete Aggregate |
|---|---|---|
| No risk of corrosion or breakage X0: for concrete without reinforcement or built-in metal elements: all influences except alternating freezing and thawing or chemically aggressive environments for concrete with reinforcement or with built-in metal inserts in a very dry environment | Concrete inside buildings with very low air humidity; foundation concrete without reinforcement in an environment without the influence of frost. | Yes |
| Corrosion of reinforcement due to carbonation Suppose concrete containing reinforcement or other embedded metal elements is exposed to air and moisture. In that case, the degree of environmental influence is determined as follows: Note Moisture conditions refer to the situation inside the cover layer of reinforcement or other embedded metal elements, but in many cases, the conditions in the cover layer can be considered the same as in the surrounding environment. In such cases, it may be appropriate to determine the impact according to the surrounding environment but not if the concrete is separated from the surrounding environment (e.g., by an insulating layer). XC1: dry or constantly wet, XC2: wet, occasionally dry, XC3/4: moderately wet, wet/alternately wet and dry | XC1: concrete inside buildings with low humidity (30–60%) air; concrete permanently immersed in water; XC2: concrete surface exposed to long-term exposure to water or high air humidity; parts of water reservoirs; most basic elements of buildings; indoor spaces with high air humidity (e.g., kitchens for mass catering, bathrooms, large laundries, areas of indoor swimming pools and barns) XC3: concrete inside buildings with medium air humidity; external concrete protected against rain; parts of buildings to which outside air often or constantly have access (e.g., open halls) XC4: concrete surfaces in contact with water, which are not included in the degree of environmental influence XC2 and XC3; parts of buildings directly exposed to precipitation or moisture | |
| Corrosion of reinforcement due to chlorides, but not from seawater Suppose concrete with reinforcement or other built-in elements comes into contact with water containing chlorides, including de-icing agents, except seawater. In that case, the environmental impact must be graded as follows: XD1: medium wet, humid | Concrete surfaces exposed to chlorides dispersed in the air; separate garages | Maybe (if exams prove suitability) |
| Corrosion of reinforcement due to chlorides from seawater If concrete with reinforcement or other built-in elements comes into contact with chlorides from seawater or salty air from seawater, then the environmental impact must be graded as follows: XS1: exposed to salty air but not in direct contact with seawater | Constructions near the sea coast or on the coast | Maybe (if exams prove suitability) |
| Alternating action of freezing and thawing with or without deicing agents If wet concrete is exposed to freezing and thawing (frost cycles), the environmental impact must be graded as follows: XF1: slightly saturated with water without deicing agents | External vertical parts of buildings exposed to rain and frost (facades of buildings, columns), not too wet parts of buildings | Yes |
| Alternating action of freezing and thawing with or without deicing agents If wet concrete is exposed to freezing and thawing (frost cycles), the environmental impact must be graded as follows: XF2: slightly saturated with water with deicing agents XF3: heavily saturated with water without deicing agents XF4: heavily saturated with water with the deicing agents | XF2: external vertical parts of buildings exposed to frost and deicing agents dispersed in the air, which are not directly in contact with the spraying of deicing agents (e.g., parts of anti-noise walls, retaining walls), and which are not included in XF4 XF3: external parts of buildings exposed to rain and frost; external parts of buildings often wet with water and exposed to frost; open water tanks; parts of structures with fluctuating freshwater levels; overflow parts of water structures XF4: parts of buildings directly exposed to deicing agents and frost; building structures near roads exposed to direct spraying with deicing agents (e.g., curbs, drainage gutters); reservoirs near roads, concrete barriers | Maybe (if exams prove suitability) |

The disadvantage of using recycled aggregates to remove phosphates from wastewater is the increased pH of the water above 11. If the water were cleaned with recycled material, it would be necessary to neutralize the water before discharge. Scientists from Britain are using recycled concrete aggregate to produce concrete pavers. When creating parts, it is necessary to increase the proportion of fine aggregate to coarse aggregate in a ratio of 4 to 1 due to the final surface layer. A high replacement percentage has a common harmful effect, but for tensile strength, a full replacement of up to % of both coarse and fine fractions is recommended. The main disadvantage of using recycled concrete aggregate is the increased proportion of water. An increased proportion of water adversely affects frost resistance and, therefore, the product's overall life. Therefore, it is necessary to consider the amount of water used during production as a decisive aspect of using recycled aggregate. Recycled concrete is used together with slag aggregate in the production of asphalt mixtures [40–43].

Recycled aggregate is used in the construction sector primarily as a base layer (68%) and as aggregate for asphalt mixtures (9%). At about 7% of use, aggregate is used as a component or backfill material. Moreover, 6% of recycled concrete aggregate is used as a substitute for aggregate in concrete, and approx. 3% is used as backfill in the vicinity of water bodies (water management sector) (Figure 3).

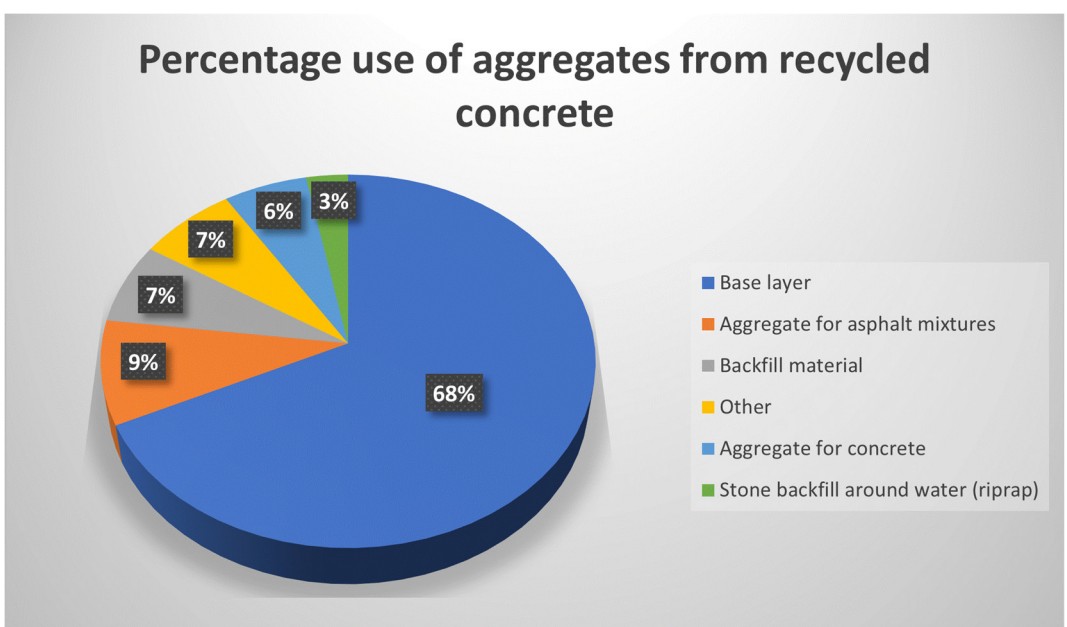

**Figure 3.** Percentage use of aggregates from recycled concrete.

### 3. Results and Discussion

Already during the design and construction of the building, we should think about its demolition potential and use materials that are easily recyclable and do not burden the environment. At the same time, we should avoid harmful varnishes and coatings that disrupt circular construction. Contaminated material (for example, wood impregnated with harmful substances) makes it impossible to recycle the product.

The CEd BIM (circular economic dimension of BIM) parameter represents an innovative parameter or dimension that combines three commonly used dimensions of information modeling, namely dimensions 5D: costs, 6D: sustainability and 7D: facility management (Figure 4).

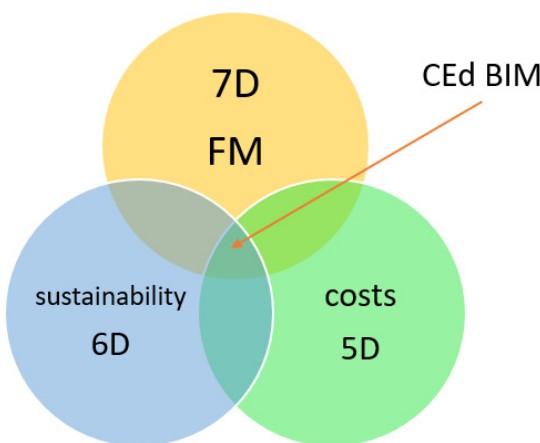

**Figure 4.** CEd BIM.

The mentioned parameter provides the user with comprehensive information about procurement costs (input costs), sustainability (information about the carbon footprint) and the life cycle of the product. Input costs are among the key indicators. Budgeting (setting the price) is an important part of the product life cycle. The main task is to determine the amount of cost necessary for production, including profit. Determining the level and amount of costs is a demanding process that considers the quality of the product and, at the same time, the impacts on selected aspects of sustainability—economic, environmental and social aspects. An important part of the assessment of products and elements is the consideration of energy efficiency during the product's life cycle. Recycled materials are important in supporting the sustainable life cycle of materials and products. The possibility of recycling materials and their reuse represents an important aspect and a step towards promoting a circular economy and sustainable management. Products and products will be through CEd BIM; I bring a comprehensive picture of the impacts of the mentioned element on the circular economy. The final classification of products whose primary raw material and component is concrete, or recycled aggregate, according to the newly proposed CEd BIM parameter. The final classification of products contains a list of all products and products that can be produced and used as concrete recyclate products or aggregates. Each element contains the CEd BIM parameter level—its graphic and verbal description. The higher the degree of fulfilment of the goals of the circular economy, the more suitable, or more favorable, the element can be considered from the point of view of sustainable use and support of the circular economy.

The result of the implemented research was the design of a BIM library of sustainable elements made from recycled concrete aggregate. The BIM library links manufacturers with BIM digital replicas of real building products and components. The innovative parameter CEd BIM (circular economic dimension of BIM) will be designed in the library, which provides the user with basic information about the impact on costs and aspects of sustainability (carbon footprint data) for products and products using recycled concrete aggregate. As part of the habilitation work, a parameter structure was proposed considering three dimensions of building information modeling—5D: costs, 6D: sustainability and 7D: facility management. The goal is to implement the mentioned parameters in the information modeling of buildings, respectively, in the newly developed application Detailer. The application will provide the user with visual information (through a three-dimensional model) and non-graphical information—structural or material specifications, information on the need for financial resources and information on the impact on selected aspects of the circular economy. The elements defined in this way will carry similar information and data as the elements created in the building information modeling environment. The goal is to integrate the designed parameter, or library of elements, into the environment of building information modeling by exporting the output of the designed database in IFC format, which is the standard output of work in building information modeling and virtual

reality. Subsequently, such a model will be able to be extracted into various BIM software applications and visualized through virtual reality, which will support the visualisation of construction production. The investor will get a detailed picture of his future work, and suppliers and subcontractors will be able to plan better and manage the production process.

Recycled concrete aggregate will be used as a substitute for natural aggregate in the implementation and construction of the following structural elements (Figure 5):

- aggregate replacement for reinforced concrete footings—made of reinforced concrete class C 16/20, steel reinforcement B 500,
- aggregate replacement for reinforced concrete piles—made of reinforced concrete class C 16/20, steel reinforcement B 500,
- aggregate replacement for reinforced concrete columns (400 × 400 mm)—made of reinforced concrete class C 30/37, steel reinforcement B 500-exposed concrete,
- aggregate replacement for reinforced concrete walls—made of reinforced concrete class C 30/37, steel reinforcement B 500-exposed concrete,
- aggregate replacement for reinforced concrete prefabricated channels (400 × 400 mm)—made of reinforced concrete class C 30/37, steel reinforcement B 500-exposed concrete,
- aggregate replacement for reinforced concrete ceiling slabs—made from reinforced concrete class C 16/20, steel reinforcement B 500-exposed concrete,
- aggregate replacement for the construction of reinforced concrete stairs—made of reinforced concrete class C 30/37, steel reinforcement B 500,
- underfill under the sidewalks,
- underfill under asphalt/concrete surfaces (underground parking lot and driveway),
- filling of sewers,
- creation of drainage layers.

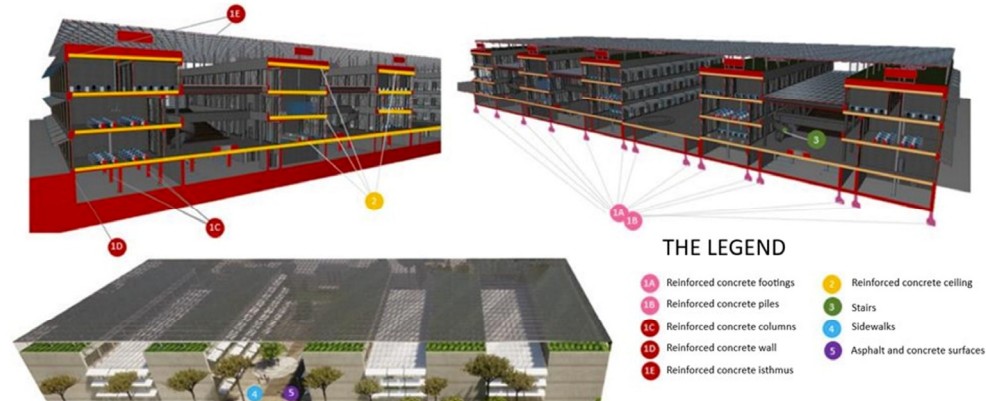

**Figure 5.** Visualization—location of the object.

The priority of the construction of the centre is to build a sustainable building, i.e., to implement the work using sustainable methods with the greatest possible use of sustainable materials and procedures, which will reduce the impact on the ecosystem and support the goals of a circular economy. Traditional, natural raw materials will be replaced by recycled, so-called secondary raw materials within individual constructions and elements.

The foundation structure consists of a system of concrete footings and piles. The concrete of the foundation footings and piles is made of reinforced concrete of class C 16/20 with steel reinforcement B 500. The total area of reinforced concrete required to realize reinforced concrete footings and piles is approximately 20 m$^3$. As part of the research, we analyzed variant solutions for the realization of foundation footings and piles from natural aggregates and recycled concrete aggregates (Figure 6). In the case of using natural aggregates, the values of the selected parameters—carbon footprint ($CO_2$ emissions), input costs (LCC) and the CEd BIM parameter—are more unfavorable than in the case of the use of recycled concrete aggregates. Although the input costs are at the same level (level 2, medium input costs), from the point of view of the carbon footprint, i.e., the number of $CO_2$

emissions that burden the environment, recycled concrete aggregates show more favorable values, namely level 2, the medium value of $CO_2$ emissions (recycled aggregate—input material concrete without admixtures and fittings, sorted by the recycling unit), respectively, 1st level—the low value of $CO_2$ emissions for recycled aggregate, where the input material is concrete without admixtures and fittings, sorted separately. From the point of view of the CEd BIM parameter, the optimal choice of aggregate for the concrete of foundation footings and piles is the replacement of aggregate from recycled concrete without admixtures and fittings, where sorting takes place separately, without a sorting unit.

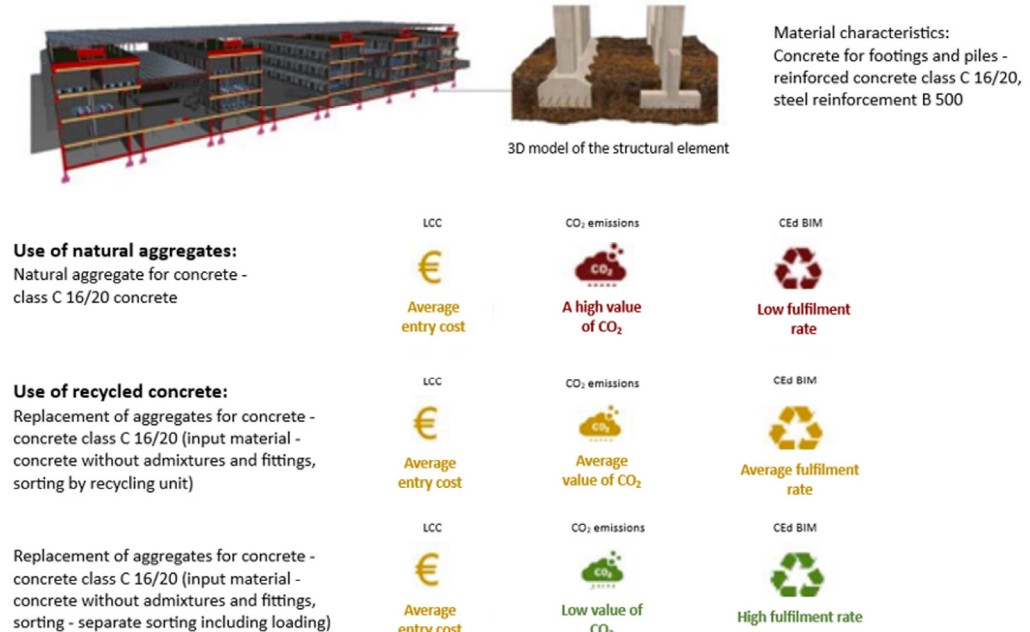

**Figure 6.** Vertical and horizontal load-bearing structures—columns, walls, ceiling slabs, diesel.

## 4. Conclusions

In connection with the worldwide strengthening of initiatives aimed at changing the global economic and social environment from linear to circular, the need to create, process and use information and knowledge about the sustainability of every economic activity, including the processing of materials and products at the end of their life cycle, is continuously growing. And reuse in the form of recycled raw materials. Information of a technical, technological, legislative, economic, environmental and social nature forms a complex whole in its scope and meaning, which must be managed in such a way as to create added management value within the circular economy. The use of building information modeling (BIM) and knowledge systems changes the ways of planning and managing individual processes. Many software applications allow creating and managing information models containing various parameters. Despite the number of software tools, there is a lack of a unified information environment enabling complex management and management of the life cycle of buildings with elements from recycled raw materials (recycled aggregate or recycled concrete) with a connection to data and information generated by the global, national or regional circular economy in the area of the use of sustainable building materials. The main output of the research was, based on the gradual fulfillment of the goals at the theoretical and practical level, the proposal of the concept of the use and processing of information about recycled raw materials in such a way as to enable their effective use in the construction industry, namely in the planning process through an effective tool—information modeling. Data and information about available products from recycled raw materials (aggregate, recycled concrete) are integrated with the database (BIM library). They can be used in the information model of the building during the construction preparation phase. The database makes it possible to define the priority properties of

products with elements of recycled raw materials through several parameters, following the goals of the circular economy. By creating a concept of using this integrated information about the products in question, we identified a common intersection between individual dimensions of information modeling containing information about costs, sustainability and management of constructions (buildings) during the life cycle. It was thus possible to connect the concept of using modern information technologies to increase the potential of using non-energy recycled raw materials and materials to support the circular economy and sustainable development.

**Author Contributions:** Conceptualization, M.B. and A.B.; methodology, M.B. and A.B.; software, M.B. and A.B.; validation, M.B. and A.B.; formal analysis, M.B. and A.B.; investigation, M.B. and A.B.; resources, M.B. and A.B.; data curation, M.B. and A.B.; writing—original draft preparation, M.B. and A.B.; writing—review and editing, M.B. and A.B.; visualization, M.B. and A.B.; supervision, M.B. and A.B.; project administration, M.B. and A.B.; funding acquisition, M.B. and A.B. All authors have read and agreed to the published version of the manuscript.

**Funding:** This research received no external funding.

**Institutional Review Board Statement:** Not applicable.

**Informed Consent Statement:** Not applicable.

**Data Availability Statement:** No applicable.

**Conflicts of Interest:** The authors declare no conflict of interest.

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
