# Peer review of "Advanced Innovation Technology of BIM in a Circular Economy"

_applsci, doi:10.3390/app13137989_

Round 1

Reviewer 1 Report

This article discusses the advanced innovative technologies of BIM in the circular economy, the main problems in this paper are as follows:

1. What are the innovative points of this paper, which should be clearly presented in the abstract and introduction.

2. The clarity of the pictures in this article needs to be improved.

3. The references cited in this paper are old, it is suggested to quote more references in the past five years.

4. The format of this article still has some problems and needs to be modified.

5. Overall, the author of this paper has done a good job and recommends that it is accepted after minor revision.

Author Response

Thank you very much for your positive review, rating, and valuable time devoted to our manuscript. I greatly appreciate your stimulating suggestions and recommendations to increase the level of our manuscript. I believe that the modifications made by us will lead to your satisfaction.

Reviewer 2 Report

Summary of paper and its contributions: the paper proposes the application of BIM for Circular Construction. 

Comments:

1) The theoretical and practical contributions of this paper is borderline. Overall, the paper is quite truncated and I am unable to see what is the knowledge gap, what is the objective of the study, what is the research design and methodology, what are the key findings and how has it been validated?

2) There should be more literature review of BIM for Sustainability.

3) Perhaps the authors can also look into how the concept of Material Passport can be a bridge/ connector between Circular Economy and BIM. 

Specific comments: 

1) The title is inappropriate. BIM should not be considered as "Advanced Innovation/ Technology". The concept of BIM has been around since the 1970s. Also, "Circular Construction" would be more suitable than "Circular Economy" in the context of the paper. 

2) The authors should not use first person narrative such as "I" and "my research". 

3) There are some misnomers or terms which should be explained. E.g. Line 54 "to-var". 

4) Line 55 - it should be "Spotify sells listening licenses instead of selling CDs." 

5) To rephrase Line 90 - "These raw materials are at high risk of jeopardising supplies". 

6) To rephrase Line 92 - "The list of these raw......, and last, in 2020.......". 

7) Consistency in font. Line 110 - "Construction raw materials" should be bolded and in italics as with "industrial minerals" and "metallic minerals". 

8) There should be explanation of "industrial minerals" as with "construction raw materials". 

9) Line 173 - "Bim" should be in capital letters, i.e. "BIM". 

10) Line 173 - use another, commonly accepted definition of BIM. 

11) Quality/ resolution of Figure 2-6 are poor and should be replaced. In addition, can authors confirm these images were created by the authors themselves and not sourced, and should be referenced? 

12) Table 1 can be better aligned. Table header can either be centralised or aligned from the left. 

13) Line 272 - What is CEd? 

14) There is no comment on useability in 2nd item "Corrosion of reinforcement due to carbonation" of Table 1. In other items, there would be comments on useability such as Yes/ Maybe. In addition, the remarks "exams prove suitability" which has been used 3 times, should be rephrased and explained. 

15) Some references, such as #28, are not suitable. For #28, please replace with more seminal and academic publications instead of sourcing/ referencing a private company. 

pls refer to above. 

Author Response

Thank you very much for your review and the valuable time devoted to our manuscript. The review is very detailed; it is clear, clear and, above all, concrete, which greatly facilitated the subsequent editing of the manuscript, for which we thank you once again.

We greatly appreciate your stimulating suggestions, recommendations and comments to improve the level of our manuscript. I believe that the modifications made by us will lead to your satisfaction.

The only comment - no. 2, we could not incorporate, as the second reviewer reproached us for writing in the plural. So it is questionable whether writing the manuscript in the singular or the plural is correct...

Reviewer 3 Report

The aim of this paper is the proposal of the concept of the use and processing of information about recycled raw materials in such a way as to enable their effective use in the construction industry, namely in the planning process through an effective tool - information modelling. It was thus possible to connect the concept of using modern information technologies to increase the potential of using non-energy recycled raw materials and materials to support the circular economy and sustainable development.

The research design is appropriate. The methodology is explained. The title of the paper "Advanced Innovation Technology of BIM in a Circular Economy" reflects its content. Possible future tasks described.

Listed below are some comments

Very poor quality illustrations Figure 2., Figure 5. and Figure 6. Absolutely unreadable.

I don't feel qualified to judge about the English language and style, but I pay attention to:

1 . Abstract: Based on my research, my results will be presented... although the authors is 2 people

2. wherefrom " habilitation work" in line 301 The result of the habilitation work ...

Author Response

Thank you very much for your positive review, rating, and valuable time devoted to our manuscript. I greatly appreciate your stimulating suggestions and recommendations to increase the level of our manuscript. I believe that the modifications made by us will lead to your satisfaction.

The only comment - no. 1, we could not incorporate, as the second reviewer reproached us for writing in the singular. So it is questionable whether writing the manuscript in the singular or the plural is correct...

Round 2

Reviewer 2 Report

The authors only partially addressed some of the issues and incorporated some of the recommendations. 

Minor revisions. 

Author Response

Again, thank you very much for your review and the valuable time you have devoted to our manuscript after editing it in the first round of reviews.
We have incorporated your comment about "modern" BIM technology. We supplemented the manuscript with the historical development of BIM and, based on this, we deleted the word modern, even though in most countries of the world, this technology is modern, as they are just becoming familiar with it and starting to apply it.
I believe that our adjustments will already lead to your satisfaction.
